# High-resolution mapping of the neutralizing and binding specificities of polyclonal sera post-HIV Env trimer vaccination

Adam S Dingens[1]*, Payal Pratap[2], Keara Malone[1], Sarah K Hilton[1], Thomas Ketas[3], Christopher A Cottrell[2], Julie Overbaugh[4], John P Moore[3], PJ Klasse[3], Andrew B Ward[2], Jesse D Bloom[1,5]*

[1]Basic Sciences Division and Computational Biology Program, Fred Hutchinson Cancer Research Center, Seattle, United States; [2]Department of Integrative Structural and Computational Biology, The Scripps Research Institute, La Jolla, United States; [3]Department of Microbiology and Immunology, Weill Cornell Medical College, New York, United States; [4]Human Biology Division, Fred Hutchinson Cancer Research Center, Seattle, United States; [5]Howard Hughes Medical Institute, Seattle, United States

**Abstract** Mapping polyclonal serum responses is critical to rational vaccine design. However, most high-resolution mapping approaches involve isolating and characterizing individual antibodies, which incompletely defines the polyclonal response. Here we use two complementary approaches to directly map the specificities of the neutralizing and binding antibodies of polyclonal anti-HIV-1 sera from rabbits immunized with BG505 Env SOSIP trimers. We used mutational antigenic profiling to determine how all mutations in Env affected viral neutralization and electron microscopy polyclonal epitope mapping (EMPEM) to directly visualize serum Fabs bound to Env trimers. The dominant neutralizing specificities were generally only a subset of the more diverse binding specificities. Additional differences between binding and neutralization reflected antigenicity differences between virus and soluble Env trimer. Furthermore, we refined residue-level epitope specificity directly from sera, revealing subtle differences across sera. Together, mutational antigenic profiling and EMPEM yield a holistic view of the binding and neutralizing specificity of polyclonal sera.

*For correspondence: adingens@fredhutch.org (ASD); jbloom@fredhutch.org (JDB)

## Introduction

Mapping polyclonal antibody responses is central to understanding antigen-specific humoral immunity. However, it is difficult to disentangle the multiple epitope specificities within polyclonal responses. Often, serum neutralization assays or ELISAs with variant antigens are used to crudely map epitope specificities. But these and other traditional serological mapping approaches do not provide high-resolution, residue-level information for the multiple components of polyclonal serum responses. Cloning and characterizing many individual monoclonal antibodies (*Scheid et al., 2009*) have revolutionized our understanding of serum responses, but antibody cloning can be biased by the isolation strategy, is not proportional to antibody serum abundance or potency, and fails to characterize the entirety of the serum neutralization response. Further advances in understanding polyclonal sera have been made through techniques that rely on high-throughput B-cell receptor sequencing (*Kreer et al., 2020*), mass spectrometry-based approaches to directly sequence

**eLife digest** Vaccines work by stimulating the immune system to produce proteins called antibodies. These antibodies bind to the virus targeted by the vaccine and block the virus from infecting cells. It has been difficult to develop a vaccine for HIV because frequent mutations allow it to evade antibodies. Understanding exactly how these proteins bind to HIV and how various mutations enable the virus to escape them is crucial to designing a successful HIV vaccine.

Over the last decade, scientists have developed new techniques for studying individual antibodies and how they bind to viruses. Now, they are using these insights to design vaccines. Most vaccines result in the production of many antibodies that bind to different parts of the virus, making it harder for a virus to escape. But studying many antibodies with different targets on the virus simultaneously remains challenging.

By combining two-cutting edge approaches, Dingens et al. catalogued the many antibodies that rabbits produce in response to an experimental vaccine for HIV. In the experiments, they mapped how two types of rabbit antibodies target the virus: those that could bind to the virus, and those that could both bind and neutralize the virus (i.e., block it from infecting cells). The experiments showed that small differences between the HIV virus and the vaccine explained why some rabbit antibodies created in response to the vaccine could bind but not neutralize the virus. Moreover, the ability to stop HIV from infecting the cells appeared to be reserved to antibodies that could bind to several different locations at the virus. Dingens et al. further documented all the virus mutations that would allow it to evade neutralizing antibodies.

The techniques used in the experiments may help scientists identify the best sites on the HIV virus to target with vaccines and to better understand the binding and neutralizing activity of antibodies. The results of the experiments may also help to redesign the experimental HIV vaccine – which is currently being tested in humans – to be even more effective.

antibody proteins (*Lavinder et al., 2014*; *Wine et al., 2013*), or decomposing bulk serum-level measurements (*Ackerman et al., 2017*; *Chung et al., 2015*; *Georgiev et al., 2013*).

Only recently have techniques been developed that directly measure the antibody specificity in polyclonal sera. The first of these techniques, electron microscopy polyclonal epitope mapping (EMPEM), directly images serum Fabs bound to an antigen of interest (*Barnes et al., 2020*; *Bianchi et al., 2018*; *Boyoglu-Barnum et al., 2020*). However, this approach characterizes the binding response, whereas it is the neutralizing antibody response that is most directly correlated with vaccine protection. Here we combine EMPEM with a second complementary technique, mutational antigenic profiling (*Dingens et al., 2017*), that quantifies the effect of all single amino-acid mutations to a viral entry protein on escape from serum neutralization.

For the purpose of this study, we mapped polyclonal anti-HIV antibody responses elicited with stabilized recombinant SOSIP Env trimers. These trimers have been used extensively as immunogens because they recapitulate the native or near-native structure of Env on the virus surface (*Julien et al., 2013*; *Lyumkis et al., 2013*; *Pancera et al., 2014*; *Sanders et al., 2013*; *Sanders and Moore, 2017*). In general, immunizing animals with prototypical SOSIP trimer variants based on the BG505 strain (*Wu et al., 2006*) induces autologous, tier-2 neutralizing antibody responses (*de Taeye et al., 2015*; *Klasse et al., 2016*; *Sanders et al., 2015*; *Torrents de la Peña et al., 2018*, *Torrents de la Peña et al., 2017*). While such immunizations can protect against infection of simian–human immunodeficiency virus, bearing the matched BG505 Env in macaques (*Pauthner et al., 2017*; *Pauthner et al., 2019*), heterologous breadth has not been consistently achieved. This lack of breadth highlights the need to understand the targets of vaccine-elicited neutralizing antibodies and re-focus responses to more broadly conserved epitopes.

Prior mapping of SOSIP trimer-induced antibody responses in animal models has revealed viral strain- and species-specific hierarchical responses, with BG505 trimer immunogenicity in rabbits serving as a well-characterized model system. BG505-induced rabbit neutralizing antibodies predominantly target a BG505-specific glycan hole (GH) in Env's glycan shield, centered on the broadly conserved glycosylation sites at residues 241 and 289 that are missing in BG505. Reintroducing glycans back in at these sites eliminates much of the neutralizing activity in many rabbit serum

responses (*Klasse et al., 2018*; *Klasse et al., 2016*) and mAbs isolated from immunized rabbits target this immunodominant GH (*McCoy et al., 2016*). Serum neutralization assays with large panels of pseudovirus point mutants identified a second frequently immunogenic site in rabbits as the C3/V5 epitope (previously termed C3/465), as well as a less immunodominant and less commonly targeted epitope in V1 (*Klasse et al., 2018*). These rabbit immunogenicity data were largely corroborated using EMPEM to directly visualize serum Fabs bound to BG505 trimer bait (*Bianchi et al., 2018*). In SOSIP trimer-vaccinated guinea pigs and non-human primates, antibody cloning or EMPEM has identified additional strain-specific responses to the C3/V4, C3/V5, V1, and gp120/gp41 interface regions (*Cottrell et al., 2020*; *Lei et al., 2019*; *Nogal et al., 2020*; *Nogal et al., 2019*). Additionally, trimer immunization often elicits non-neutralizing responses to the base of the trimer, a neo-epitope exposed on soluble trimers but inaccessible on viral membrane-bound Env (*Bianchi et al., 2018*; *Cottrell et al., 2020*; *Hu et al., 2015*; *Kulp et al., 2017*). Ongoing clinical trials will evaluate the immunogenicity of BG505 SOSIP trimer variants in humans (ClinicalTrials.gov Identifiers: NCT03699241, NCT04177355, and NCT03783130).

Below, we use mutational antigenic profiling to directly map the dominant neutralizing antibody specificities present within a panel of polyclonal sera from rabbits immunized the BG505 SOSIP trimer variants. In parallel, we map the binding specificities of these sera using EMPEM, providing a holistic view of both serum binding and neutralization.

## Results

### Rabbit sera panel

We chose a small panel of rabbit sera to optimize mutational antigenic profiling of polyclonal sera. We used sera from rabbits sequentially vaccinated with BG505 SOSIP trimer variants: either BG505 SOSIP.664, which contains the T332N mutation (*Sanders et al., 2013*), or the further stabilized BG505 SOSIP.V4.1 (*de Taeye et al., 2015*), each administered either three or four times. Details of the immunization schemes and characterization of some of these rabbits' sera responses at earlier time points have been reported previously (*Klasse et al., 2018*; *Klasse et al., 2016*; *Ringe et al., 2019*; *Ringe et al., 2017*). We chose serum samples with various specificities, including sera that predominantly target the 241/289 GH or C3/V5 epitopes alone, both of these epitopes, or neither of these epitopes. To identify such sera, we performed preliminary TZM-bl neutralization assay mapping using pseudoviruses bearing mutations that affect each of these epitopes, as well as the V1 epitope rarely targeted in rabbits. The resulting sera panel and associated preliminary mapping data are shown in *Figure 1A*. These sera do not represent an unbiased collection of rabbit immune responses, but rather a curated selection of potent responses with different specificities.

### Mutational antigenic profiling of rabbit sera

We performed mutational antigenic profiling (*Dingens et al., 2017*) of each serum using libraries of replication-competent HIV virions expressing all mutants of the BG505.T332N Env (*Haddox et al., 2018*), allowing us to map autologous responses to the strain-matched BG505 trimer immunogen. Briefly, this approach (*Figure 1B*) first involves generating libraries of mutant viruses containing all single amino-acid mutations to Env compatible with viral replication. Each mutant virus library is then incubated with a highly selective concentration of sera before infecting a T-cell line such that only viruses that escape neutralization can enter cells. In our experiments, we chose serum concentrations to keep the average level of selection exerted by each serum relatively constant; across all replicates, between 0.02% and 9.27% of the library escaped neutralization, with the across-replicate averages for each serum ranging from 0.3% and 2.7% (*Figure 1—figure supplement 1*). The frequency of each mutation among viruses that are able to escape neutralization is quantified by Illumina sequencing of the viral cDNA produced in infected cells. Comparing the relative frequency of each mutation in the sera-selected condition to a non-selected control condition quantifies the effect of each mutation on resistance to sera neutralization (*Figure 1B*). As an additional control, we also incubated viral libraries with pre-vaccine sera for each rabbit. *Figure 1—figure supplement 1* details the serum dilutions, the number of replicates (three to six per post-immunization sera; median values are reported throughout), and the level of neutralization achieved by each serum in each experiment. Median mutation differential selection values across all experimental replicates for a given

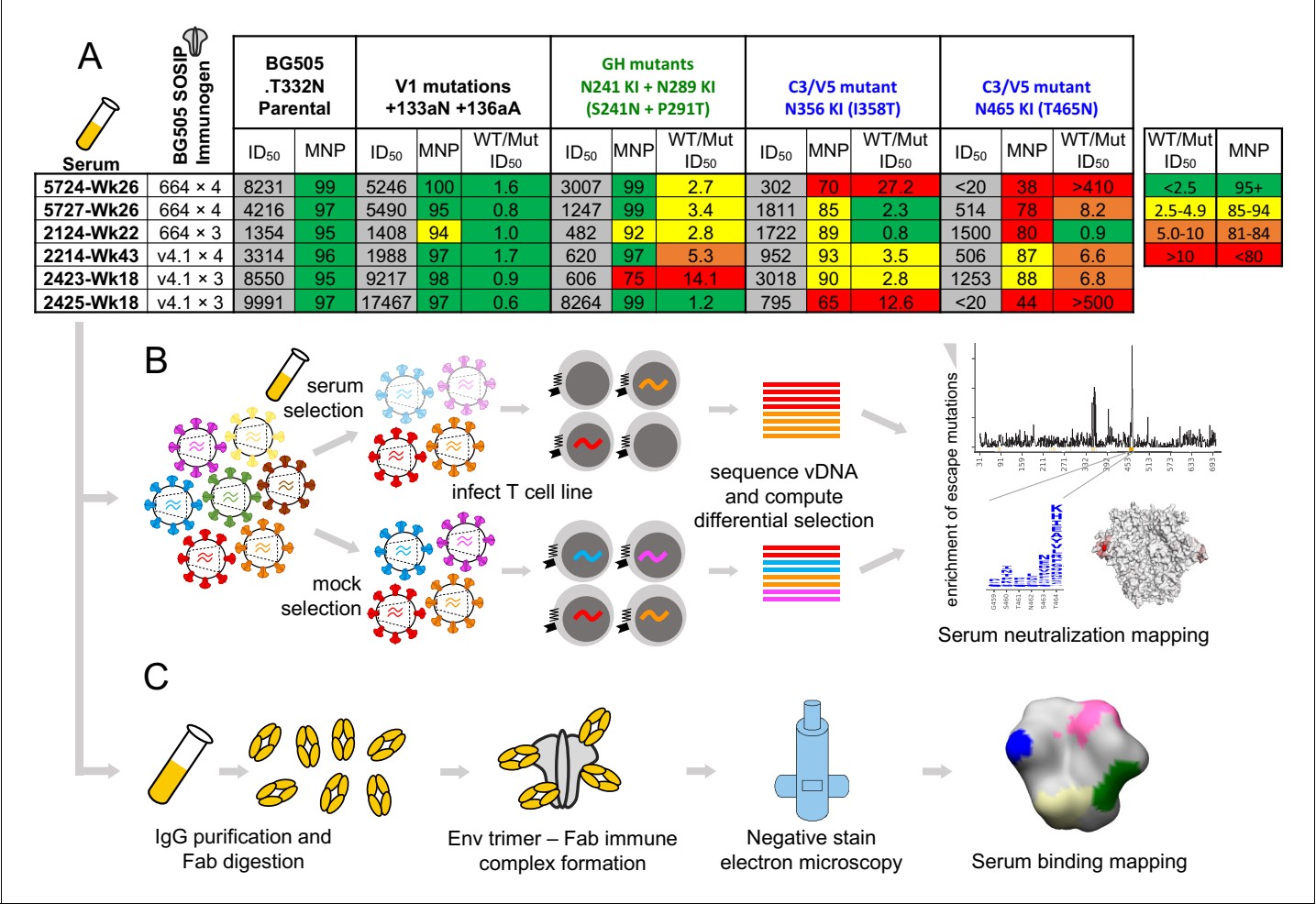

**Figure 1.** Overview of sera and epitope mapping approaches. (A) Preliminary point-mutant mapping of the BG505-trimer-vaccinated sera panel. The sera dilution that inhibits 50% of virus entry ($ID_{50}$), maximum neutralization percentage plateau (MNP), and fold change in $ID_{50}$ values of wild type relative to mutant pseudoviruses (WT/Mut $ID_{50}$) are shown for the parental BG505.T332N and pseudoviruses bearing insertion (V1 epitope) and glycan knock in mutation(s) (GH and C3/V5 epitopes). The weeks (Wk) post-initial vaccination of the serum sample is specified in each sera's name; sera are subsequently referred to by only their four-digit ID number. (B) Experimental schematic of mutational antigenic profiling. (C) Experimental schematic of EMPEM.

The online version of this article includes the following figure supplement(s) for figure 1:

**Figure supplement 1.** Experimental details for all mutational antigenic profiling experiments.

serum are presented throughout (see *Figure 1—figure supplement 1* for replicate-to-replicate correlations).

The Env mutations that affect neutralization by each serum are plotted in *Figure 2*. The results are largely concordant with prior knowledge on BG505 trimer immunogenicity in rabbits, with each serum targeting one or both of the C3/V5 or GH epitopes, which are indicated by blue and green, respectively, in *Figure 2*. Additionally, the antigenic profiling largely agrees with the crude epitope specificities mapped using a small panel of pseudovirus point mutants (*Figure 1A*, with mutations tested in preliminary TZM-bl assays plotted in black in *Figure 2B*). Notably, most sera select neutralization-escape mutations in the C3/V5 epitope to some extent, including sera from rabbit 2124, which appeared to predominantly target the GH based on the preliminary point-mutant mapping (*Figure 1A*). *Figure 2* is just one approach to visualizing these complex datasets; to facilitate more flexible data exploration, antigenic profiling data for each sera can be interactively explored in dms-view (*Hilton et al., 2020*) by visiting https://jbloomlab.github.io/Vacc_Rabbit_Sera_MAP/. This interactive visualization reduces potential interpretation and presentation biases by allowing users to

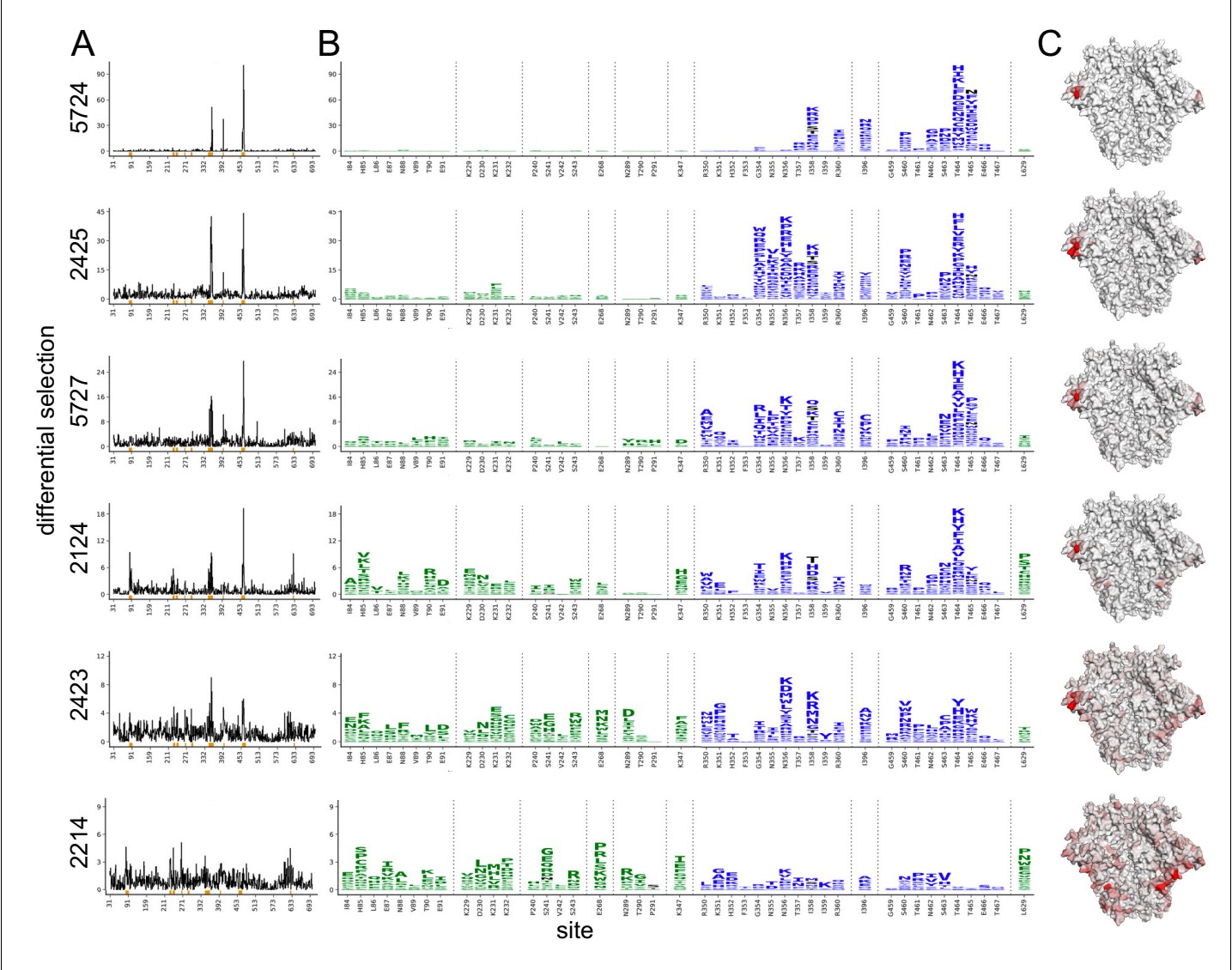

**Figure 2.** Serum neutralization-escape mutations mapped by mutational antigenic profiling. (**A**) Line plots showing the positive site differential selection for each serum across the Env ectodomain. Differential selection is a measure of the enrichment of mutations in the sera-selected conditions relative to a no-sera control (see Materials and methods for details). (**B**) Differential selection for each mutation at key sites (indicated with orange underlines in (**A**)). The height of each letter is proportional to the differential selection for that amino-acid mutation. The GH epitope is colored green, and the C3/V5 epitope is colored blue. Mutations tested during preliminary point-mutant mapping are colored black (*Figure 1A*; all mutations that add a tested glycan are indicated). For both (**A**) and (**B**), the y-axis is scaled to the maximal effect size site for each sera; sera without a single dominant region of escape (2243 and 2214) are plotted such that 90% of the site-level signal is at <20% of the y-axis maximum. (**C**) The positive site differential selection is mapped onto the BG505 trimer (PDB: 5FYL). The color scheme is right censored at 15 to visualize subdominant responses; for samples with a maximum selection less than 15, the maximum value is mapped to most red color. An interactive version of these visualizations is at https://jbloomlab.github.io/Vacc_Rabbit_Sera_MAP/. Raw numerical values and logo plots for the entire Env ectodomain are in *Figure 2—source data 1*.

The online version of this article includes the following source data and figure supplement(s) for figure 2:

**Source data 1.** Zip file containing csv files with all median site- and mutation-level differential selection values, as well as logo plots plotting escape profiles for the entire mutagenized portion of *env*.

**Figure supplement 1.** Mutational antigenic profiling data plotted for each individual replicates of pre- and post-immunization sera from rabbit 5724.

**Figure supplement 2.** Mutational antigenic profiling data plotted for each individual replicates of pre- and post-immunization sera from rabbit 2214.

**Figure supplement 3.** Mutational antigenic profiling data plotted for each individual replicates of pre- and post-immunization sera from rabbit 5727.

**Figure supplement 4.** Mutational antigenic profiling data plotted for each individual replicates of pre- and post-immunization sera from rabbit 2124.

**Figure supplement 5.** Mutational antigenic profiling data plotted for each individual replicates of pre- and post-immunization sera from rabbit 2423.

**Figure supplement 6.** Mutational antigenic profiling data plotted for each individual replicates of pre- and post-immunization sera from rabbit 2425.

**Figure supplement 7.** Validation of mutational antigenic profiling in neutralization assays.

examine the extent of neutralization-escape by mutations at any site and projecting the mutations onto interactive structures of the Env trimer or monomer.

We hypothesized that sera with a few dominant sites of escape (e.g., serum 5724 in *Figure 2*) had neutralizing activity that was strongly focused on one epitope of Env, whereas sera with smaller-effect escape mutations to multiple regions (e.g., serum 2214 in *Figure 2*) had neutralizing responses that targeted multiple distinct epitopes. To explore this hypothesis, we tested if the small effect sizes observed in mutational antigenic profiling for some sera accurately reflect the effect of mutations in TZM-bl neutralization assays. We identified the most selected mutation at the most selected site for each serum, generated pseudoviruses bearing these mutations, and tested them in serum neutralization assays. The fold enrichment in mutational antigenic profiling was well correlated with the fold change in $ID_{50}$ in the neutralization assays (*Figure 2—figure supplement 7*, Pearson's r = 0.97, p value = 0.0011 from a two-tailed Pearson correlation test). For example, the 5724 escape profile is focused entirely on the C3/V5 epitope: T464H is enriched ~190-fold in this serum's mutational antigenic profiling and shifts the $ID_{50}$ 64-fold in a TZM-bl neutralization assay (*Figure 2—figure supplement 7*). In contrast, 2423 targets both the C3/V5 and GH epitopes (*Figure 2*), and the maximal effect mutant N356K has just a ≈2-fold effect in both mutational antigenic profiling and neutralization assays (*Figure 2—figure supplement 7*). A caveat is that the extent of mutant enrichment upon serum selection is also influenced by the serum dilution used in experiments, as shown in *Figure 2—figure supplement 1–6*. However, the good correlation of the extent of focusing in the escape-mutation mapping and the TZM-bl neutralization assays suggests the mutational antigenic profiling data captures the amount of focusing in the neutralization response reasonably well.

## Contrasting the neutralizing and binding specificities of the polyclonal sera

To contrast the serum neutralization specificities described above with the serum binding specificities, we performed EMPEM on the same set of sera to directly visualize antibody binding to Env (*Figure 1C*). We reasoned that collecting both types of data would enable us to compare and contrast the sites where antibodies bind to the sites where mutations mediate escape from neutralization. *Figure 3* presents the refined 3D reconstructions from negative stain electron microscopy of serum Fabs bound to immunogen-matched BG505 SOSIP trimer alongside mutational antigenic profiling data. Across all sera, it is immediately clear that the binding responses identified by EMPEM include many epitopes where mutations do not affect viral neutralization. For five of six sera, binding responses to both the GH and C3/V5 epitopes are observed. All sera contain additional binding responses to other epitopes. For example, even 5724, where we observe narrow, strongly focused viral escape in just the C3/V5 epitope, contains numerous additional binding responses, including the GH, N611 glycan, base-of-trimer, and V1/V3 epitopes.

Some of the differences in binding vs neutralization-escape are easily explained by antigenicity differences between stabilized Env trimers and replication-competent virus. First, while the base-of-trimer epitope is presented on recombinant trimers and commonly elicited by trimer immunization (*Bianchi et al., 2018*; *Cottrell et al., 2020*; *Hu et al., 2015*), it is inaccessible in the replication-competent virus and hence does not elicit neutralizing antibodies. Therefore, base-binding responses are observed in all sera, but of course are not mapped as neutralizing epitopes. Second, binding responses to the N611 glycan region are also observed in all sera (*Figure 3*), but we do not observe viral escape by disrupting this glycosylation motif (*Figure 2*). It has been previously shown that site N611 is less glycosylated in BG505 SOSIP trimers than in virus, which enhances the immunogenicity of the exposed region when this glycan is missing (*Derking et al., 2020*). Accordingly, EMPEM identified binding responses to this region using the immunogen-matched trimer bait, while neutralizing responses are not apparent in mutational antigenic profiling because the virus libraries are likely to have higher glycan occupancy at N611. Third, gp120 interface responses are observed in two sera; both of these rabbits were immunized with BG505 SOSIP.v4 trimers. This gp120 interface epitope includes the A316W stabilizing mutation added to the SOSIP.v4 to reduce the exposure of the V3 loop (*de Taeye et al., 2015*). We have recently found that the A316W mutation alters immunogenicity to this region, eliciting mutation-specific responses to this region that would not cross-react with the A316-bearing viral libraries (manuscript in preparation). Together, these trimer-binding but non-neutralizing responses to the base, N611 glycan, and gp120 interface reflect and expand on

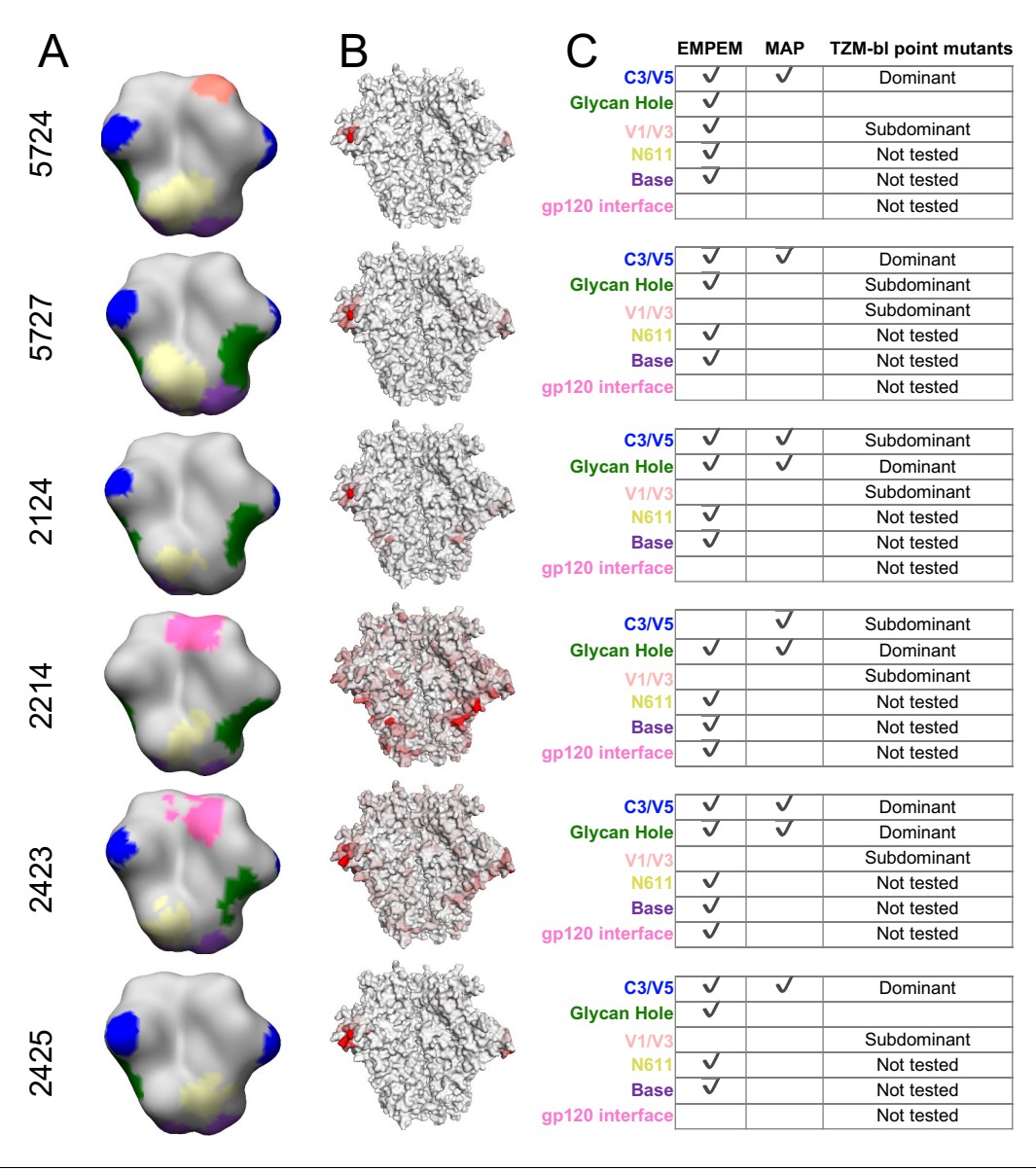

**Figure 3.** Comparing the binding and neutralization specificities of the sera. (A) Refined 3D reconstructions from negative stain EMPEM. Specificities are mapped onto the Env trimer structure, with epitopes colored as in (C). (B) Mutational antigenic profiling data mapped onto the BG505 SOSIP Env structure, as in *Figure 2C*, represented here to contrast with EMPEM. (C) Summary data from (A) and (B), as well as validation TZM-bl neutralization assay point-mutant mapping of single and double epitope knock out mutations (see *Figure 3—figure supplement 1* for details). A checkmark (EMPEM and MAP) indicates a response was detected to that epitope, and TZM-bl responses were summarized as 'Dominant', 'Subdominant', or undetected (left blank) based on data in *Figure 3— figure supplement 1*. We did not perform neutralization assay validation at epitopes where the discrepancies between the EMPEM and mutational antigenic profiling are easily explained by antigenicity differences between soluble trimer and virus (N611, base, and gp120 interface; labeled as 'Not tested').

The online version of this article includes the following figure supplement(s) for figure 3:

**Figure supplement 1.** Effect of mutations that disrupt the C3/V5, glycan hole, and V1 epitopes alone and in combination with other epitope knockouts.

previously characterized antigenic differences between Env trimers to Env on the surface of virus (*Bianchi et al., 2018*; *Cottrell et al., 2020*; *Derking et al., 2020*; *Hu et al., 2015*).

The few remaining differences in binding and neutralization could reflect a number of factors. Some binding responses may be non-neutralizing, although it is often assumed that antibodies that bind native Env present on the virus – mimicked by stabilized Env trimers – are neutralizing (*Burton et al., 2000*; *Yang et al., 2006*). Alternatively, at the serum concentrations we tested, the antibody occupancy on the virions or the binding kinetics may disfavor neutralization. Furthermore, some observed binding responses may be neutralizing but at such subdominant levels that they do not exert selective pressure in the mutational antigenic profiling at the serum concentrations tested. Alternatively, some epitopes may require saturation in order to be neutralizing and did not reach these saturating levels at the serum concentrations tested in mutational antigenic profiling.

To investigate these hypotheses, we tested each sera's ability to neutralize pseudovirus point mutants bearing mutations to one or both of the C3/V5 and GH epitopes (*Figure 3—figure supplement 1* and summarized in *Figure 3C*). Comparing the effect of single epitope mutations to multiple epitope mutations can help define the relative dominance of different neutralizing specificities (*Klasse et al., 2018*). A caveat is that neutralization assays for different sets of mutants were performed in two different laboratories; we therefore only present general interpretations in *Figure 3C*. For both 5724 and 2425, single or multiple glycan knock in mutations to the GH epitope had negligible effects, while single or multiple glycan knock in mutations to the C3/V5 epitope had large effects (*Figure 3—figure supplement 1*). For these two sera, knocking in a glycan to each epitope (S241N + I358T) did not have much larger effects than the single C3 glycan knock in (I358T), suggesting these sera have limited GH neutralizing responses even after eliminating some of the C3/V5 directed neutralizing response. While this matches the mutational antigenic profiling, binding GH responses are observed in these sera (*Figure 3A*). This suggests that these GH responses are either non-neutralizing or *much* less dominant than the C3/V5 neutralizing antibody responses. Of note, while the resolution of the negative stain EMPEM in this present study limits fine epitope interpretations, GH binding responses (specifically 'GH2'-like responses) have previously been identified in non-neutralizing sera using cryo-EMPEM (*Bianchi et al., 2018*).

When the effect of a mutation to one (e.g., epitope A) is apparent when another epitope (e.g., epitope B) is knocked out, but not when testing the epitope A mutation alone, we can interpret the neutralizing antibody response to epitope A as being 'subdominant' relative to the epitope B response. Here, 5727, 2124, 2214, and 2423 all displayed a greater effect for the double epitope glycan knock in mutations (S241N + I358T) than either of the single epitope knock in mutations. Comparing the effects of the single and double epitope mutations suggests sera 2423 had relatively equivalent neutralizing responses to both the GH and C3/V5 epitopes, sera 2124 and 2214 had a dominant response to GH and a subdominant response to C3/V5, and sera 5727 had a dominant response to C3/V5 and a subdominant response to GH (*Figure 3*, *Figure 3—figure supplement 1*). EMPEM identified binding responses to both epitopes in three of four of these sera, while mutational antigenic profiling identified neutralizing responses to both epitopes in only two of four sera (*Figure 3*).

Since there were a number of instances in which subdominant neutralizing responses to the GH or C3/V5 epitope identified with TZM-bl point-mutant mapping did not appear in either EMPEM or mutational antigenic profiling (e.g., GH response for 5727 is absent in mutational antigenic profiling, and a C3/V5 response for 2214 is absent in EMPEM), we examined if there were additional unobserved subdominant responses. We focused on the effect of insertion mutations to V1, an epitope occasionally targeted in rabbits (*Klasse et al., 2018*). We tested sera for neutralization of V1 and V1 + GH epitope mutants and compared effects to GH mutations alone. While V1 insertions alone did not affect neutralization of any sera, they had an additional effect when tested with GH mutations relative to GH mutations alone (*Figure 3—figure supplement 1*). This suggests that there are subdominant neutralizing antibody responses to V1 in all sera – though we cannot rule out that the V1 + GH double mutants broadly affect antigenicity. Mutational antigenic profiling did not clearly identify any V1 responses, while EMPEM identified a V1/V3 binding response – which overlaps the V1 insertion mutations – in one serum (5724, *Figure 3*).

## Residue-level refinement of sera epitopes

The mutational antigenic profiling data also allows for residue-level refinement of the dominant targets of the neutralizing antibody response. For example, it is immediately apparent that the clustered, surface-exposed sites 464 and 356 and/or 358 'anchor' the C3/V5 epitope, with many mutations at these sites having large effects for nearly all sera that strongly target this epitope (*Figure 4*). However, detailed epitope specificity at other C3/V5 sites varies across rabbits: some sera are more focused on various regions of C3, including sites 350 and 351 (e.g., sera 5727, 2423, 2124) or 354–357 (e.g., sera 5727, 2423, 2124, 2425), whereas serum 5724 is more narrowly focused on the V5 region previously identified by knocking in a glycan at site 465. Notably, site 396 bridges these two epitope regions in both linear sequence and structural space and is a site of escape for most sera.

We validated this residue-level specificity using TZM-bl neutralization assays (*Figure 4*). For example, mutations to sites 350, 351, 355, and 356 have very little effect on 5724, while mutations to 358 and 464 have large effects. Similarly, 2425 is most focused on residues 354–358 in the C3 region, mutations to these sites gave larger effects, while those to 350 and 251 did not. While the magnitude of effect size varied across sera (note the differing y-axis in *Figure 4*), the TZM-bl mapping generally reflected these effect sizes.

It is also clear that the mutational antigenic profiling better explores the effects of different possible escape mutations than testing smaller panels of pseudovirus mutants. For example, while knocking in a glycosylation site at site 465 (T465N) escapes many of the sera as previously shown (*Klasse et al., 2018*), other mutants at this site have similar or even larger effects (*Figures 2* and *3*). This suggests that the immune response is directed at site 465 and neighboring V5 residues, as opposed to targeting this general epitope region that is obstructed by adding a bulky glycan to site 465.

There is greater variation in both the residue-level specificity and magnitude of neutralizing antibody responses to the GH epitope. However, overlaying data onto the trimer structures make it clear that a subset of the sera target the 241/289 GH. Examining data for three sera with clear enrichment in this region reveal selective pressure in a large epitope region, spanning from site 85 near the fusion peptide, through the 241/289 GH region, and extending to site 347 in/near the C3/V5 epitope. While we arbitrarily classify site 347 as part of the GH epitope because enrichment at this site more closely tracks with sera that target the GH epitope (*Figure 2*), the blending of these epitopes supports the notion that the C3/V5 and GH epitopes together constitute a BG505-specific immunodominant 'super-epitope' in rabbits (*Nogal et al., 2020*). Mutations at site 629, particularly L629P, are also enriched strongly in two sera that target the GH epitope (*Figure 5*). Site 629 is near the N-terminus of the HR-2, located below and buried beneath the GH epitope in the pre-fusion structure; the proline mutation may disrupt the HR-2 alpha helix, altering its antigenicity or accessibility of the GH epitope.

Mutations in the GH epitope were oftentimes only moderately enriched compared to the C3/V5 epitope, and TZM-bl assays also showed only small shifts in the neutralization curve for these mutants. Sera without clear enrichment of escape mutations in this epitope also were not affected by epitope mutations in TZM-bl neutralization assays (*Figure 5—figure supplement 1*).

## Discussion

We have used mutational antigenic profiling to map the dominant neutralizing antibody specificities in polyclonal rabbit sera elicited with Env trimer immunization. In parallel, we used EMPEM to map the total serum binding specificities. Contrasting the serum binding and neutralizing specificities suggests that the dominant neutralizing antibody responses are only a subset of binding responses – even when just examining bona fide neutralizing antibody epitopes. Additional differences in binding and neutralization highlight antigenicity differences between soluble SOSIP Env trimers and Env on the surface of viruses, which is relevant to the use of trimers as immunogens.

This work also shows the utility of mutational antigenic profiling in mapping polyclonal serum responses to HIV Env. We recapitulate and extend prior knowledge on rabbit antibody responses to BG505 trimer immunization (*Bianchi et al., 2018*; *Klasse et al., 2018*; *McCoy et al., 2016*), with the C3/V5 and GH regions being the dominant immunogenic epitopes. We map responses to multiple epitopes of polyclonal serum responses at once, a significant advance over more traditional mapping

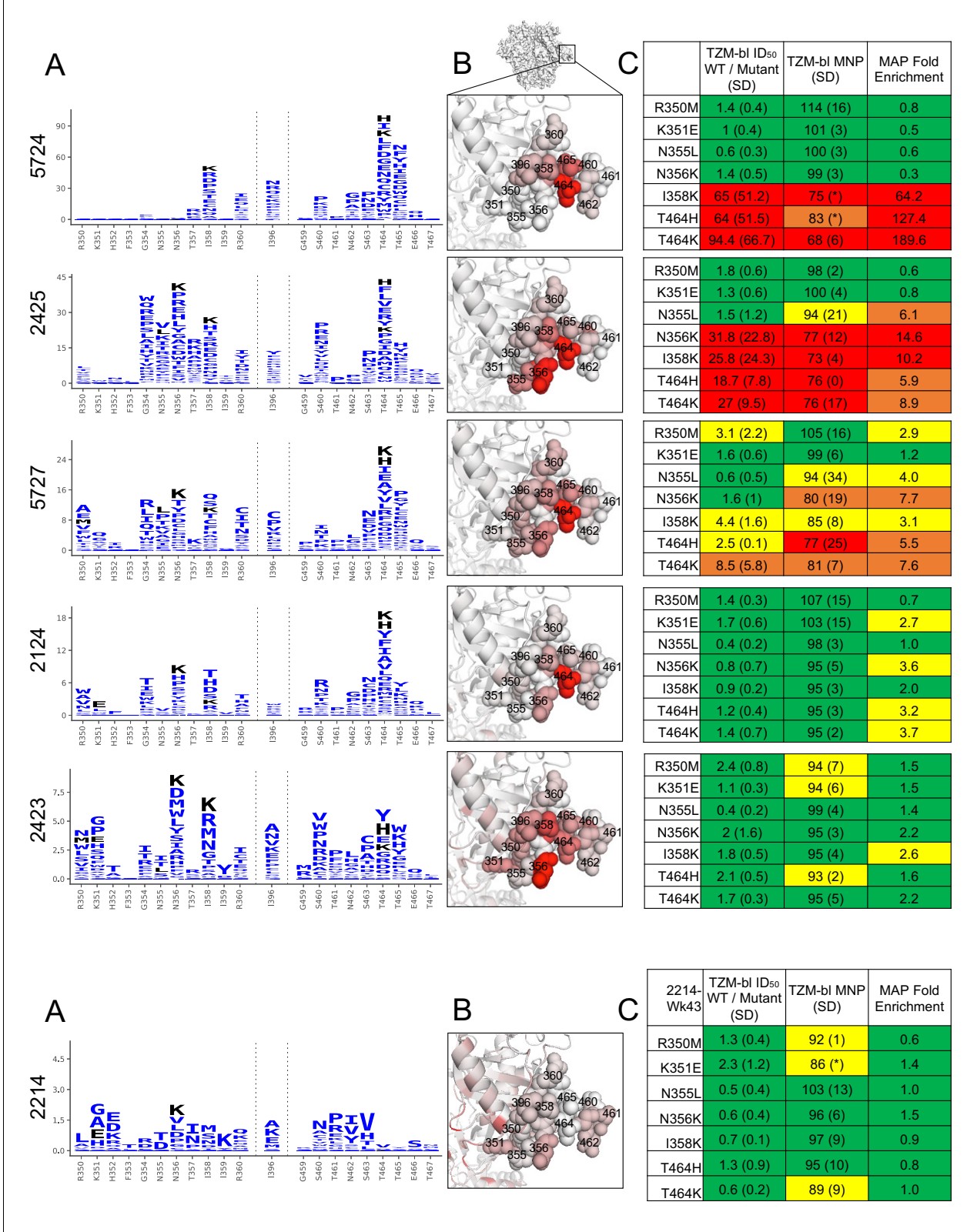

**Figure 4.** C3/V5 residue-level epitope specificity. (A) Differential selection is plotted in logo plots for the C3/V5 epitope, as in **Figure 1B**. Mutations validated in TZM-bl neutralization assays (C) are colored black. The y-axis is scaled to the largest effect size site. (B) The positive site differential selection is mapped onto the BG505 trimer (PDB: 5FYL), colored as in **Figure 1C**. Sites shown in (A) are shown with spheres in (B). These data can be explored interactively using dms-view at https://jbloomlab.github.io/Vacc_Rabbit_Sera_MAP/. (C) The fold change in ID50 relative to wild type and the
*Figure 4 continued on next page*

*Figure 4 continued*

maximum neutralization plateau (MNP) for each mutant validated in a pseudovirus TZM-bl neutralization assay. The table color scheme is as in *Figure 1A*, and the standard deviation of replicate measures are shown in parentheses. See *Figure 4—figure supplement 1* for the single sera with relatively less targeting of this epitope.

The online version of this article includes the following figure supplement(s) for figure 4:

**Figure supplement 1.** A, B, and C as in *Figure 4*, but for the single sera with limited targeting of the C3/V5 epitope.

approaches. Furthermore, we refine the residue-level specificity of these epitopes directly from sera, revealing fine-grain epitope differences. For example, some C3/V5 responses were more focused on the C3 region than others. But there were sites that appeared to 'anchor' this epitope – all sera targeting this epitope were strongly affected by mutations to sites 358 and 454. It remains to be determined if these different epitope specificities are the result of monoclonal responses with varying

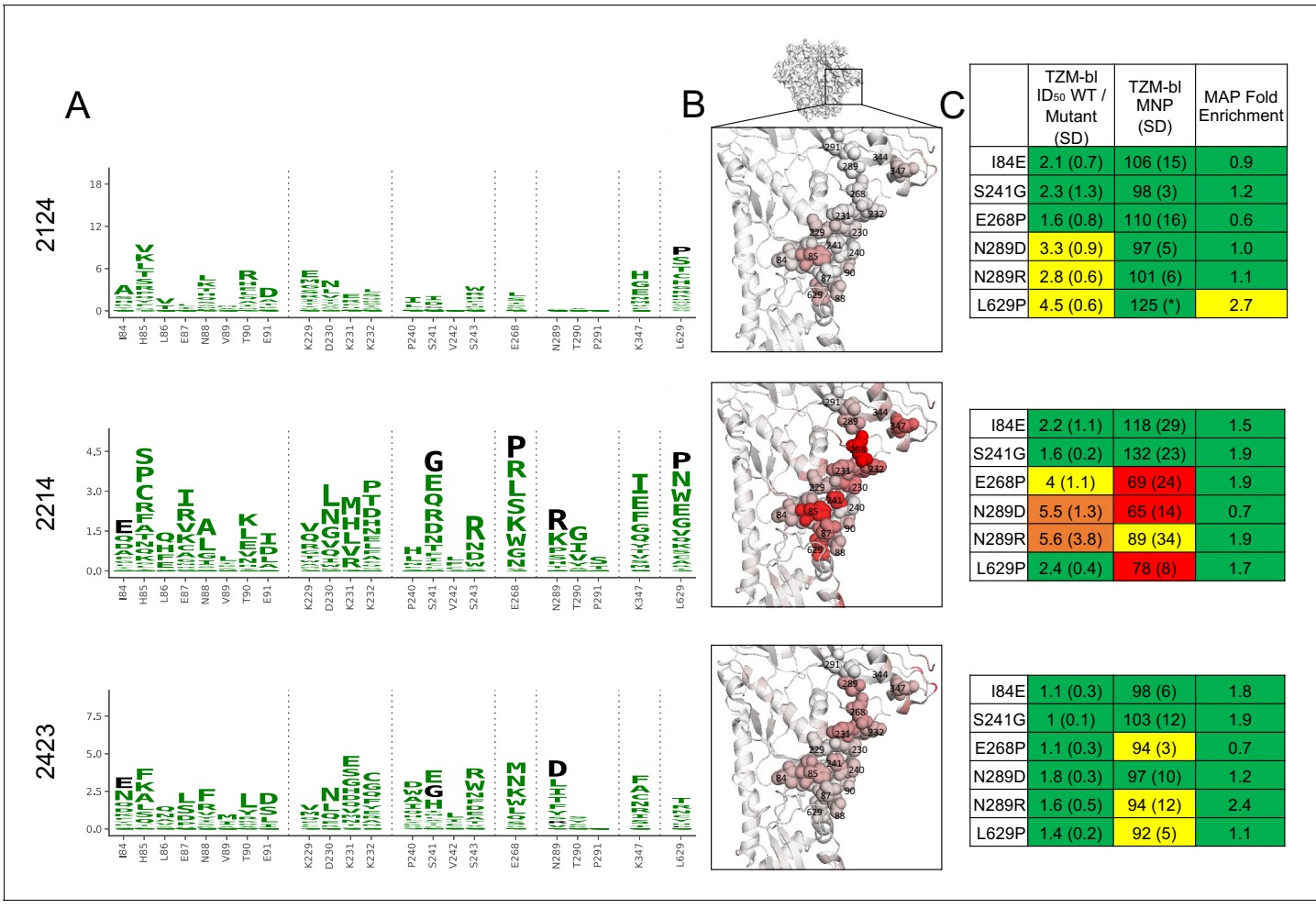

**Figure 5.** Glycan hole epitope specificity. (A) Differential selection is plotted in logo plots for the glycan hole epitope, as in *Figure 1B*. Mutations validated in TZM-bl neutralization assays (C) are colored black. The y-axis is scaled to the largest effect size site, which is oftentimes not in this GH view. (B) The positive site differential selection is mapped onto the BG505 trimer (PDB: 5FYL), with a single monomer shown and colored as in *Figure 1C*. Sites shown in (A) are shown with spheres in (B). These data can be explored interactively using dms-view at https://jbloomlab.github.io/Vacc_Rabbit_Sera_MAP/. (C) The fold change in ID50 relative to wild type and the maximum neutralization plateau (MNP) for each mutant validated in a pseudovirus TZM-bl neutralization assay. The table color scheme is as in *Figure 1A*, and the standard deviation of replicate measures are shown in parentheses. The table color scheme is as in *Figure 1A*. See *Figure 5—figure supplement 1* for the three sera with limited targeting of this epitope.

The online version of this article includes the following figure supplement(s) for figure 5:

**Figure supplement 1.** (A), (B), and (C) as in *Figure 5*, but for the three sera with limited targeting of the glycan hole epitope.

binding footprints or if they vary due to differing sets of multiple overlapping antibodies targeting the same epitope region. Lastly, while rabbit antibody responses to BG505 SOSIP trimer immunization have been well-characterized, these unbiased sera mapping approaches have the potential to identify novel specificities. Indeed, some sera do display minor enrichment of possible escape mutations outside of known epitopes (e.g., sites 507 and 509 for sera 5727).

The GH epitope is not a desirable broadly neutralizing epitope that can be exploited for vaccine design (*Yang et al., 2020*), and the C3/V5 epitope is similarly problematic due to its relatively low sequence conservation. While the autologous neutralizing responses we mapped here are encouraging, we hypothesize it will be important to silence these immunodominant and autologous neutralizing responses to aid in redirecting response to more broadly conserved epitopes. This is particularly important given that similar specificities have also been observed in non-human primates after BG505 SOSIP trimer immunization (*Cottrell et al., 2020*; *Nogal et al., 2020*). Our mutational antigenic profiling provides a rich map of potential alterations that disrupt these epitopes, which could be of use in resurfacing trimer immunogens.

While this work has allowed us to compare total polyclonal serum neutralization and binding, there are important limitations to keep in mind. One key point is that these methods have different sensitivities for dominant versus subdominant responses, making it impossible to directly quantify serum neutralization and binding on the same scale. Indeed, TZM-bl neutralization assays using single and double epitope mutant pseudoviruses suggest that both approaches may have 'missed' subdominant neutralizing antibody responses to the V1 epitopes (*Figure 3*, *Figure 3—figure supplement 1*). The mutant libraries used to map neutralization have on average only ~1 Env mutation per virus; a given virion can therefore likely only escape one component of polyclonal responses, limiting the ability to detect subdominant responses. Furthermore, rigorous quantification of both binding and neutralizing responses – and comparisons of responses across sera – are not yet possible with existing technology and analytical approaches. Furthermore, while mutational antigenic profiling examines the entire serum neutralizing antibody response, EMPEM maps the binding of IgG Fabs after purification. Lastly, easily visualizing and interpreting these complex datasets remains difficult, a challenge we have attempted to overcome here with the interactive visualizations available at https://jbloomlab.github.io/Vacc_Rabbit_Sera_MAP/. Nonetheless, this work describes the most detailed mapping yet of the specificities of polyclonal sera at both the binding and neutralizing level.

While both rational, structure-based vaccine design (*Alam et al., 2017*; *Correia et al., 2014*; *Dubrovskaya et al., 2019*; *Haynes et al., 2012*; *Kong et al., 2019*; *Kwong and Mascola, 2018*; *Saunders et al., 2017*; *Xu et al., 2018*) and broadly neutralizing antibody germline-targeting (*Briney et al., 2016*; *Dosenovic et al., 2015*; *Escolano et al., 2019*; *Escolano et al., 2016*; *Jardine et al., 2013*; *Jardine et al., 2016*; *Medina-Ramírez et al., 2017*; *Steichen et al., 2019*; *Steichen et al., 2016*; *Tian et al., 2016*) approaches have begun to show exciting promise for HIV, continued progress will require iterative rounds of evaluating vaccine responses and redesigning immunogens and vaccine regimens (*Ward and Wilson, 2020*). Determining the extent of immunofocusing to targeted epitopes, while also tracking – and subsequently eliminating – off-target responses to less conserved regions will be critical to these efforts. Together, EMPEM and mutational antigenic profiling can aid in rational vaccine design by directly mapping polyclonal serum specificities for both binding and neutralization.

# Materials and methods

## Mutational antigenic profiling

Mutational antigenic profiling was performed as previously described (*Dingens et al., 2017*). Briefly, $5 \times 10^5$ infectious units of BG505.T332N mutant virus libraries (*Haddox et al., 2018*) were neutralized with serum dilutions for 1 hr at 37°C as specified in *Figure 1—figure supplement 1*. We performed additional experimental replicates for sera with smaller effect sizes in order to increase our signal relative to noise (*Figure 1—figure supplement 1*). All replicates (*Figure 1—figure supplement 1*) were included in the analysis, without any removal or quality filtering of whole replicates or individual mutation counts. Different library numbers (e.g., 1, 2, 3) are viral libraries generated from independently generated DNA libraries (*Haddox et al., 2018*), and letter labels (e.g., a, b, c)

correspond to experiments done on different days with matched non-selected control library selections. After neutralization, viral libraries were then infected into $1 \times 10^6$ SupT1.CCR5 cells in R10 containing 100 µg/mL DEAE-dextran. Three hours post-infection, the cells were resuspended in 1 mL R10 (RPMI [GE Healthcare Life Sciences; SH30255.01], supplemented with 10% fetal bovine serum, 1% 200 mM L-glutamine containing a 1% of a solution of 10,000 units/mL penicillin and 10,000 mg/mL streptomycin). At 12 hr post-infection, cells were washed once with phosphate-buffered saline, and non-integrated viral cDNA was isolated from cells using a miniprep. Each mutant virus library was also subjected to a mock selection (no serum), and duplicate four 10-fold serial dilutions of each mutant virus library were also infected into $1 \times 10^6$ cells to serve as an infectivity standard curve (*Figure 1—figure supplement 1*). The proportion of the library that survived neutralization and entered cells was quantified using a qPCR and interpolation of the infectivity standard curve (*Dingens et al., 2019*). Sequencing libraries were generated using a barcoded subamplicon sequencing approach as previously described (*Haddox et al., 2018*; *Haddox et al., 2016*) and detailed at https://jbloomlab.github.io/dms_tools2/bcsubamp.html. Libraries were sequenced using $2 \times 250$ bp paired-end Illumina HiSeq runs.

Data was analyzed with dms_tools2 version 2.2.6 (https://jbloomlab.github.io/dms_tools2/) (*Bloom, 2015*). Differential selection is the log2-transformed enrichment of a given mutation relative to wild type in the sera-selected condition relative to the non-selected control condition. This statistic is corrected for sequencing error, determined by sequencing wild-type plasmid, in a site- and mutation-specific manner. To reduce noise associated with low sequencing counts for a given mutations, a pseudocount of 5 (scaled up for the deeper-sequenced sample by the relative sequencing depth of the selected and non-selected libraries, to avoid biases introduced by differing sequencing depths across samples) is added to sequencing counts. See prior work (*Dingens et al., 2017*; *Doud et al., 2017*) or https://jbloomlab.github.io/dms_tools2/diffsel.html for additional details.

Short tandem repeat profiling on our stock of SupT1.CCR5 cells found that 11 of 14 alleles plus both amelogenin alleles matched the reference of the parental SupT1 cells (ATCC #CRL-1942) reference profile. These cells also tested negative for mycoplasma.

## Electron microscopy polyclonal epitope mapping

IgG from rabbit sera was affinity-purified with equal parts of Protein G Sepharose (Sigma–Aldrich, P3296) and 150 ml Protein A Sepharose (GE Healthcare, 17-5138-01) in a Poly-Prep Chromatography Column (Bio-Rad, 731–1550). Fabs were generated by Papain cleavage and purified by the use of the Pierce Fab Preparation Kit according to the manufacturers' instructions (Thermo Scientific - #44985). The purity of the Fabs was confirmed by SDS–PAGE.

BG505 SOSIP.664 or BG505 SOSIP v4.1/Fab complexes were made by mixing 15 µg SOSIP with 1 mg of polyclonal Fabs and allowed to incubate for 18–24 hr at room temperature. Complex samples were SEC purified using a Superose 6 Increase 10/300 GL (GE Healthcare) column to remove excess Fab prior to electron microscopy grid preparation. Fractions containing the SOSIP/Fab complexes were pooled and concentrated using 10 kDa Amicon spin concentrators (Millipore). Samples were diluted to 0.02 mg/mL in TBS (0.05 M Tris pH 7.4, 0.15 M NaCl) and adsorbed onto glow discharged carbon-coated Cu400 EM grids (Electron Microscopy Sciences) and blotted after 10 s. The grids were then stained with 3 µL of 2% (wt/vol) uranyl formate, immediately blotted, waiting for 10 s before being stained again for 35 s followed by a final blot. Image collection and data processing were performed on TFS Talos F200C microscope (1.98 Å/pixel; 73,000× magnification) with an electron dose of ~25 electrons/Å$^2$ using Leginon (*Pugach et al., 2015*; *Suloway et al., 2005*). 2D classification, 3D sorting, and 3D refinement were conducted using Relion v3.0 (*Zivanov et al., 2018*). EM density maps were visualized using UCSF Chimera (*Pettersen et al., 2004*) and segmented using Segger (*Pintilie et al., 2010*). Figures were generated using UCSF Chimera (*Pettersen et al., 2004*).

## TZM-bl neutralization assay

TZM-bl neutralization assays used to preliminarily map the specificity of our sera panel (*Figure 1A*) were performed in laboratories at Weill Cornell as previously described (*Klasse et al., 2018*). The remainder of the TZM-bl neutralization assays were performed in laboratories at the Fred Hutch as previously described (*Dingens et al., 2017*). These protocols are very similar and based on widely used protocols such as TZM-bl neutralization assay protocols (*Sarzotti-Kelsoe et al., 2014*).

Validation assays were completed in technical duplicate two to four times. In a small number of instances, the top of the neutralization plateau did not fit for single replicates; while fold change in $ID_{50}$ values were always calculated using at minimum two replicates (with standard deviations [SD] between biological replicates reported), the maximum neutralization plateau with SD reported as '(*)' were from single replicates in which the naturalization plateau fits accurately.

## Data availability

The entire mutational antigenic profiling analysis pipeline, as well as processed data are available as https://github.com/jbloomlab/Vacc_Rabbit_Sera_MAP; *Dingens, 2021*; copy archived at swh:1:rev: b8d312d2bf5c2c117ce1d1601ea3738b58e62c20. Illumina sequencing reads were uploaded to the NCBI SRA as BioProject PRJNA656582 with sample identifiers SRR12431153-SRR12431189. EMPEM 3D maps are deposited into the EMDB with codes EMD-23366 to EMD-23371.

## Acknowledgements

We thank Caelan Radford for comments on this manuscript and the Fred Hutch Genomics Core for performing the Illumina sequencing. We thank Gargi Debnath and Erik Francomano for their assistance in purifying and digesting IgG from sera. This work was supported by the National Institute of Allergy and Infectious Diseases (NIAID) of the National Institutes of Health (NIH) grant R01 AI140891 (to JDB), P01 AI110657 (to JPM, ABW, P.J.K.), R01 AI36082 (to J.P.M, P.J.K.), and R01 AI12096 (to JO). JDB is an Investigator of the Howard Hughes Medical Institute.

## Additional information

### Competing interests

Julie Overbaugh: Reviewing editor, *eLife*. The other authors declare that no competing interests exist.

### Funding

| Funder | Grant reference number | Author |
| --- | --- | --- |
| National Institute of Allergy and Infectious Diseases | R01 AI140891 | Jesse D Bloom |
| National Institute of Allergy and Infectious Diseases | P01 AI110657 | John P Moore<br>Andrew B Ward<br>PJ Klasse |
| National Institute of Allergy and Infectious Diseases | R01 AI12096 | Julie Overbaugh |
| Howard Hughes Medical Institute | | Jesse D Bloom |
| National Institutes of Health | R01 AI36082 | PJ Klasse<br>John P Moore |

The funders had no role in study design, data collection and interpretation, or the decision to submit the work for publication.

### Author contributions

Adam S Dingens, Conceptualization, Software, Formal analysis, Investigation, Writing - original draft, Writing - review and editing; Payal Pratap, Software, Formal analysis, Investigation, Writing - review and editing; Keara Malone, Investigation; Sarah K Hilton, Data curation, Software, Visualization; Thomas Ketas, Resources, Investigation; Christopher A Cottrell, Resources; Julie Overbaugh, Conceptualization, Writing - review and editing; John P Moore, Conceptualization, Resources; PJ Klasse, Conceptualization, Resources, Writing - review and editing; Andrew B Ward, Conceptualization, Supervision, Writing - review and editing; Jesse D Bloom, Conceptualization, Software, Supervision, Writing - review and editing

## Author ORCIDs

Adam S Dingens https://orcid.org/0000-0001-9603-9409
Payal Pratap http://orcid.org/0000-0002-7170-6866
Julie Overbaugh https://orcid.org/0000-0002-0239-9444
PJ Klasse https://orcid.org/0000-0001-8222-278X
Andrew B Ward http://orcid.org/0000-0001-7153-3769
Jesse D Bloom https://orcid.org/0000-0003-1267-3408

## Decision letter and Author response

Decision letter https://doi.org/10.7554/eLife.64281.sa1
Author response https://doi.org/10.7554/eLife.64281.sa2

## Additional files

### Supplementary files

• Transparent reporting form

### Data availability

The entire mutational antigenic profiling analysis pipeline, as well as processed data are available at https://github.com/jbloomlab/Vacc_Rabbit_Sera_MAP [copy archived at https://archive.softwareheritage.org/swh:1:rev:b8d312d2bf5c2c117ce1d1601ea3738b58e62c20/]. Illumina sequencing reads were uploaded to the NCBI SRA as BioProject PRJNA656582 with sample identifiers SRR12431153-SRR12431189. EMPEM 3D maps are deposited into EMDB with codes EMD-23366 to EMD-23371. Figure 2-Source Data 1 contains csv files with all median site- and mutation-level differential selection values, as well as logoplots plotting escape profiles for the entire mutagenized portion of env.

The following dataset was generated:

| Author(s) | Year | Dataset title | Dataset URL | Database and Identifier |
|---|---|---|---|---|
| Dingens A, Bloom J | 2020 | Mutational antigenic profiling a rabbit sera responses to HIV-1 Env trimer immunization | https://www.ncbi.nlm.nih.gov/bioproject/PRJNA656582/ | NCBI BioProject, PRJNA656582 |

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
