## [Decision Letter]

**Acceptance summary:**

This paper represents a timely complementary study to map neutralizing and

binding responses in polyclonal rabbit sera induced by immunization with a native-like HIV-1 envelope trimer. These results will be instructive to researchers analyzing antibody responses to many different pathogens.

**Decision letter after peer review:**

Thank you for submitting your article "High-resolution mapping of the neutralizing and binding specificities of polyclonal sera post HIV Env trimer vaccination" for consideration by *eLife*. Your article has been reviewed by three peer reviewers, one of whom is a member of our Board of Reviewing Editors, and the evaluation has been overseen by Miles Davenport as the Senior Editor. The following individual involved in review of your submission has agreed to reveal their identity: Brandon Dekosky (Reviewer #3).

The reviewers have discussed the reviews with one another and the Reviewing Editor has drafted this decision to help you prepare a revised submission.

Summary:

Dingens et al. report a timely complementary study to map neutralizing and binding responses in polyclonal rabbit sera induced by immunization with the BG505 SOSIP Env trimer. Neutralizing responses are mapped using libraries of replication-competent HIV expressing all mutants of the BG505 Env, an approach developed in the Bloom laboratory. Binding responses were mapped using an EM-based method, EMPEM, developed in the Ward laboratory. The Env mutations that affect neutralization of the autologous BG505 strain in the BG505-SOSIP-immunized animals were largely known from other studies, as were the binding (not necessarily neutralizing) responses – the strength of this study is the combination of the two approaches. It is especially useful that the complex datasets have been deposited on-line where they can be interactively explored, including mapping onto Env trimer and monomer structures, and we commend the authors for the development of a facile and easy-to-use interactive viewer for exploring the mutational scanning data. Although the results were anticipated, it is very nice to directly compare the neutralization epitopes to the binding epitopes determined by EMPEM. This is a well-written and beautifully illustrated paper that we are happy to recommend publication after addressing the following concerns.

Essential revisions:

1) One of the main advantages of the mutational scanning approach is that it can identify novel epitopes targeted by antibody responses in a high-throughput manner. It is a little disappointing that this advantage was not leveraged in the current manuscript, perhaps due to the choice of the vaccine (BG505 SOSIP trimers where the epitopes have been thoroughly mapped in the literature) and the selection of vaccinated animals. Looking at Figure 2, animal 5727 was the only animal whose serum showed some selection signatures outside of the regions considered in depth (at sites 507 and 509) – have the authors analyzed these escape mutations? If not, and only if possible within reasonable workload, we urge the authors to pursue this example or any other example where a potential novel epitope discovery could be possible.

2) Additional information on the bioinformatic methods for data analysis is needed. How did the authors handle discrepancies in data across replicates or libraries, for example if a mutation that was enriched in one library or replicate, but deleted in another? Were there any quality filters or metrics used to estimate true signal vs. noise?

3) Differential selection statistics are mentioned briefly, along with citations to prior publications. Prior citations are definitely helpful. I think it is still important to state the key steps used in processing NGS data and the statistical techniques and quality metrics that were used. The authors should also state any criteria for acceptance or rejection or binning of individual data points, or acceptance/rejection of datasets or replicates, if quantitative criteria or metrics were used.

4) Several replicates showed a low percentage infectivity (Figure 1—figure supplement 1, e.g. animals 5724 and 2124), but the text indicates averages between 0.3% and 2.7% infectivity. Were some groups omitted from analysis, or were all groups included?

5) How well did the mutational profiles correlate between different libraries or replicates of the same samples?

---

## [Author Response]

Essential revisions:1) One of the main advantages of the mutational scanning approach is that it can identify novel epitopes targeted by antibody responses in a high-throughput manner. It is a little disappointing that this advantage was not leveraged in the current manuscript, perhaps due to the choice of the vaccine (BG505 SOSIP trimers where the epitopes have been thoroughly mapped in the literature) and the selection of vaccinated animals. Looking at Figure 2, animal 5727 was the only animal whose serum showed some selection signatures outside of the regions considered in depth (at sites 507 and 509) – have the authors analyzed these escape mutations? If not, and only if possible within reasonable workload, we urge the authors to pursue this example or any other example where a potential novel epitope discovery could be possible.

The reviewer is correct to point out that a major strength in the unbiased mapping approaches we used is their ability to potentially identify new antibody specificities. The reviewer is also likely correct that we failed to identify new antibody epitopes due to our choice of well-characterized sera from BG505 SOSIP trimer vaccinated rabbits. In this study, we chose to focus on such sera to allow for ample cross-validation not only between mutational antigenic profiling and EMPEM, but also to existing mapping data. We are excited to use these approaches to identify novel epitopes using sera that do not map to known epitopes using traditional methods in future work.

We have not tested the effect of any individual mutations (*e.g.* mutations to sites 507 and 509) not already presented in the manuscript and have limited resources to do so given COVID-19-related staffing limitations. Instead, we have added text to the Discussion on the very valid point made in this comment:

“Lastly, while rabbit antibody responses to BG505 SOSIP trimer immunization have been well-characterized, these unbiased sera mapping approaches have the potential to identify novel specificities. Indeed, some sera do display minor enrichment of possible escape mutations outside of known epitopes (e.g. sites 507 and 509 for sera 5727).”

2) Additional information on the bioinformatic methods for data analysis is needed. How did the authors handle discrepancies in data across replicates or libraries, for example if a mutation that was enriched in one library or replicate, but deleted in another? Were there any quality filters or metrics used to estimate true signal vs. noise?

The data presented are the median mutation-level enrichment statistic (differential selection) of all experimental replicates; no filtering was performed prior to taking the median value. The reason we did not perform filtering is that we do not have a statistically justified way to distinguish true signal from noise, and so felt that a simple median was the best approach. We used the median rather than the mean to avoid disproportionate effects of any single outlier point. We have made this clearer in the manuscript, adding the following sentence to the main text when discussing replicates:

“Median mutation differential selection values across all experimental replicates (Figure 1—figure supplement 1) are presented throughout.”

3) Differential selection statistics are mentioned briefly, along with citations to prior publications. Prior citations are definitely helpful. I think it is still important to state the key steps used in processing NGS data and the statistical techniques and quality metrics that were used. The authors should also state any criteria for acceptance or rejection or binning of individual data points, or acceptance/rejection of datasets or replicates, if quantitative criteria or metrics were used.

We have expanded on and added more details to the calculation of the differential selection statistic in the Materials and methods.

As described above, there was no filtering or binning of individual mutation-level enrichments; the median value was taken across all experimental replicates (results are generally unchanged if you take the mean). For this manuscript, we also did not eliminate any whole experimental replicates, including those in which we only achieved ~90% neutralization when using relatively less sera. In previous work, we have had *ad hoc* removal of experimental replicates if that replicate did not exert enough selective pressure on the library to yield selection statistics that were well correlated with other replicates (oftentimes when titrating sera/antibody). To clarify this in the manuscript, we have added the following statement in the Materials and methods:

“All replicates (Figure 1—figure supplement 1) were included in the analysis, without any removal or quality filtering of whole replicates or individual mutation counts.”

4) Several replicates showed a low percentage infectivity (Figure 1—figure supplement 1, e.g. animals 5724 and 2124), but the text indicates averages between 0.3% and 2.7% infectivity. Were some groups omitted from analysis, or were all groups included?

As described above, no experimental replicates were omitted from the manuscript or analysis. Rather, the text was just poorly written. When averaging the level of infectivity for all replicates within each serum, this serum-specific average was between 0.3% and 2.7%. Individual replicates for sera did fall outside of this range of the averages (see Figure 1—figure supplement 1). We have corrected the language to make these data clearer, and give the replicate-specific range (0.02% to 9.27%). Again, we acknowledge the basic point that some replicates are probably “less good” than others. But in the absence of a rigorous way to determine which are better, we did not feel justified in omitting any replicates. For this reason, we took the median of all replicates, reasoning that the median is less susceptible than the mean to distortion by any given replicate.

5) How well did the mutational profiles correlate between different libraries or replicates of the same samples?

The Pearson’s r correlation coefficient of the positive site differential selection values between replicates, with the different libraries used in each replicate labeled, is presented in Figure 1—figure supplement 1. The actual correlation plots at the site and mutation level are available in the linked GitHub repository, but we feel that displaying the correlation coefficient in a heat plot – as we do in the supplement – makes it much easier to digest the data than including 50+ correlation plots.